# Does a complex intervention targeting communities, health facilities and district health managers increase the utilisation of community-based child health services? A before and after study in intervention and comparison areas of Ethiopia

Della Berhanu [1,2] Yemisrach Behailu Okwaraji,[1] Atkure Defar [2,3]
Abebe Bekele,[2] Ephrem Tekle Lemango,[4] Araya Abrha Medhanyie,[5]
Muluemebet Abera Wordofa,[6] Mezgebu Yitayal,[7] Fitsum W/Gebriel,[8] Alem Desta,[5]
Fisseha Ashebir Gebregizabher,[5,9] Dawit Wolde Daka [10]
Alemayehu Hunduma,[6,11] Habtamu Beyene,[8,12] Tigist Getahun,[13,14]
Theodros Getachew,[2,13] Amare Tariku Woldemariam,[15] Desta Wolassa,[2]
Lars Åke Persson [1,2] Joanna Schellenberg[1]

For numbered affiliations see end of article.

**Correspondence to**
Dr Della Berhanu;
Della.Berhanu@lshtm.ac.uk

## ABSTRACT

**Introduction** Ethiopia successfully reduced mortality in children below 5 years of age during the past few decades, but the utilisation of child health services was still low. Optimising the Health Extension Programme was a 2-year intervention in 26 districts, focusing on community engagement, capacity strengthening of primary care workers and reinforcement of district accountability of child health services. We report the intervention's effectiveness on care utilisation for common childhood illnesses.

**Methods** We included a representative sample of 5773 households with 2874 under-five children at baseline (December 2016 to February 2017) and 10 788 households and 5639 under-five children at endline surveys (December 2018 to February 2019) in intervention and comparison areas. Health facilities were also included. We assessed the effect of the intervention using difference-in-differences analyses.

**Results** There were 31 intervention activities; many were one-off and implemented late. In eight districts, activities were interrupted for 4 months. Care-seeking for any illness in the 2 weeks before the survey for children aged 2–59 months at baseline was 58% (95% CI 47 to 68) in intervention and 49% (95% CI 39 to 60) in comparison areas. At end-line it was 39% (95% CI 32 to 45) in intervention and 34% (95% CI 27 to 41) in comparison areas (difference-in-differences −4 percentage points, adjusted OR 0.49, 95% CI 0.12 to 1.95). The intervention neither had an effect on care-seeking among sick neonates, nor on household participation in community

## Strengths and limitations of this study

► We conducted a rigorous effectiveness evaluation of an intervention to increase the utilisation of services for sick under-five children using baseline (December 2016 to February 2017) and endline (December 2018 to February 2019) surveys, in intervention and comparison areas located in 56 districts across four regions of Ethiopia.

► Data were triangulated with service utilisation records from health posts and health centres, which supported the household-level findings of no change in service utilisation as a result of the intervention.

► Although only few of the household characteristics differed between intervention and comparison areas over time and had only a marginal influence on the analyses, it is possible that unmeasured confounders might contribute to the observed results.

► At baseline, the overall proportion of reported illness in the last 2 weeks was lower than anticipated, and this was higher at the endline indicating that the difference in the reported proportion of sick children could be due to differences in the interaction and the extent of the probing between data collectors and families when enquiring about childhood illnesses.

► Data collectors were blinded to the allocation, that is, whether the district where they collected data was an intervention or comparison area, indicating that any differential reporting of care utilisation between intervention and comparison areas is, therefore, unlikely.

engagement forums, supportive supervision of primary care workers, nor on indicators of district accountability for child health services.

**Conclusion** We found no evidence to suggest that the intervention increased the utilisation of care for sick children. The lack of effect could partly be attributed to the short implementation period of a complex intervention and implementation interruption. Future funding schemes should take into consideration that complex interventions that include behaviour change may need an extended implementation period.

**Trial registration number** ISRCTN12040912.

## INTRODUCTION

In the period 1990–2015, Ethiopia reduced under-five mortality by 67%.[1] The Ethiopian Demographic and Health Surveys (DHS) showed infant mortality rates at 77, 48 and 43 deaths per 1000 live births, in 2005, 2016 and 2019 reports, respectively.[2–4] Between 2005 and 2016, neonatal mortality decreased from 39 to 29 deaths per 1000 live births, but in 2019 had stagnated at around 30. Both health system expansion and socioeconomic improvements have contributed to the reductions in mortality.[5]

In 2003, the Ethiopian government launched the health extension programme to increase primary care services.[1] Salaried female workers, known as health extension workers (HEWs), were trained to provide basic community-based services. Two HEWs at the health post serve a population of approximately 5000 people. Five health posts and their referral health centre and primary hospital comprise a primary healthcare unit. The health extension programme has 17 packages that fall under four broad areas: family health services, disease prevention and control, hygiene and environmental sanitation and health education and communication.[1] The HEWs are supported by the Women's Development Army leaders, a network of volunteer women established in 2011, who, along with other development goals, promote healthy practices in the community.[6 7]

In 2010, the integrated community case management (iCCM) strategy was launched, allowing HEWs to manage common childhood illnesses in children under 5 years of age, including pneumonia, malaria and diarrhoea.[8] In 2013, the community-based newborn care (CBNC) was integrated into the health extension programme.[9] This initiative enabled HEWs to provide antibiotics for young infants with symptoms of possible serious bacterial infection when a referral was not possible.

Care-seeking for sick under-five children has remained low.[10] In 2016, 46% of children with diarrhoeal diseases received oral rehydration therapy, and one-third of children with suspected pneumonia were taken to an appropriate care provider.[2]

Several barriers to the utilisation of services for childhood illnesses have been identified. The perceived quality of services provided by the HEWs, financial constraints and preference for alternative care providers affected the utilisation of child health services.[11–13] Service uptake was associated with maternal education levels, parents' knowledge of danger signs, the type of illness, local beliefs and the need for permission from family decision-makers to seek care.[12 14–16] Health post closure, absence of HEWs, lack of essential drugs and supplies, distance and the quality of services also contributed to the low service utilisation.[12 14 17 18] Insufficient supervision and lack of government accountability and ownership of child health services also constituted barriers.[8]

Based on a barriers-and-facilitators study, the Ethiopian Government initiated the Optimising the Health Extension Programme (OHEP) intervention.[19] OHEP was based on the following hypotheses: community engagement activities would enhance caregiver knowledge and household practices. Furthermore, capacity strengthening would improve the availability of quality services in the CBNC and iCCM programmes, and promotion of district-level ownership and accountability would integrate these services into the district-level planning and budgeting. These different components together would hypothetically lead to increased utilisation of CBNC and iCCM services.

This study aimed to assess the extent to which the OHEP intervention increased care-seeking for children under the age of 5 years, by comparing changes over time in intervention and comparison areas. Secondary outcomes included treatment of sick children with diarrhoea, suspected pneumonia or neonatal sepsis.

## METHODS

### Study setting

The Ethiopian Government initiated the OHEP intervention in 26 districts of Amhara, Southern Nation, Nationalities and Peoples, Oromia and Tigray regions with an approximate population of 3.5 million (figure 1). Intervention districts were selected by the government of Ethiopia and implementing partners for having both a relatively low utilisation of primary child health services and the availability and ability of partners to support implementation. The implementers were four nongovernmental organisations (PATH, UNICEF, Save the Children and Last 10 Kilometres/John Snow Inc.). The intervention started in 2016 and lasted for a duration of 2.5 years and had three components: 1) community engagement, 2) primary care level capacity building and 3) ownership and accountability of child health services at the district level. The intervention activities under these components, along with the underlying assumptions, intermediate indicators and outcomes are detailed in table 1.

### Study design

The protocol for the evaluation of the OHEP implementation has been published.[20] This study was based on a plausibility design with 26 intervention and 26 comparison districts (woredas) in four regions of Ethiopia. The baseline survey was conducted from December 2016 to February 2017 and the endline survey from December 2018 to February 2019 (figure 1). The surveys were

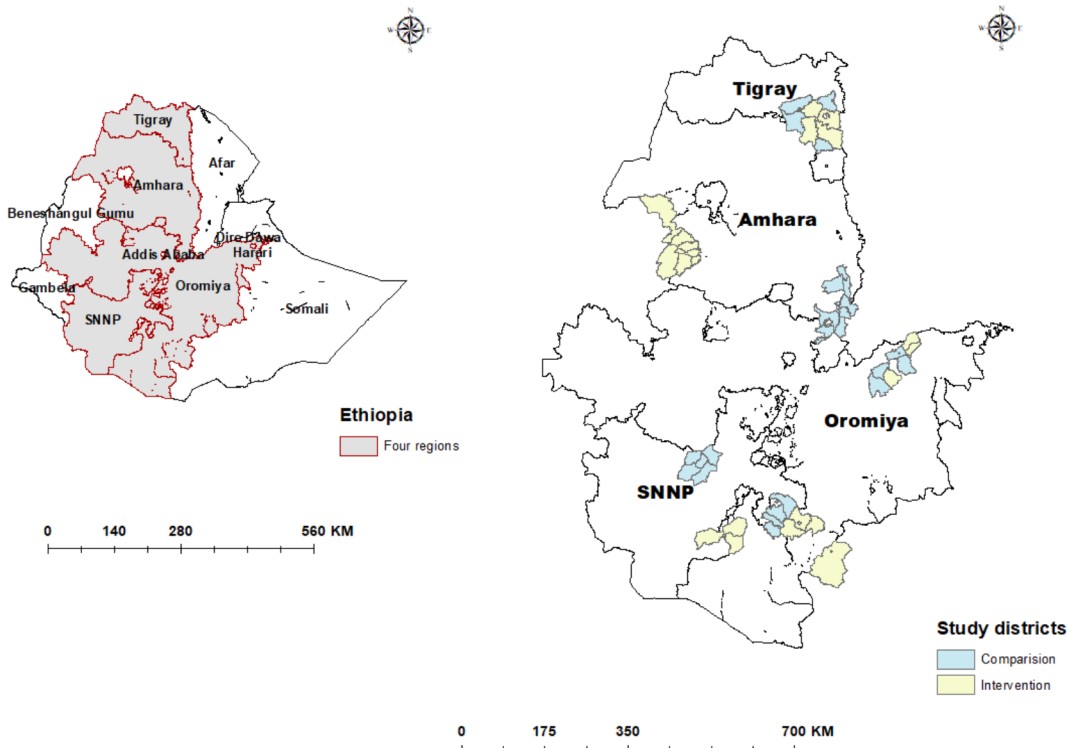

**Figure 1** Map of Ethiopia showing all regions (left) and the intervention and comparison districts within the four study regions (right).

conducted by the London School of Hygiene and Tropical Medicine and Ethiopian Public Health Institute along with representatives from Gondar, Jimma, Mekelle and Hawassa Universities.

## Sample

Intervention districts had a low coverage of maternal, newborn, and child health indicators. Comparison districts were selected by the Regional Health Bureaus to be similar to the size of the population, the burden of diseases, number of primary healthcare units, health service coverage, length of iCCM and CBNC service delivery and absence of partners implementing other demand generation activities.

We used a two-stage stratified cluster sampling to select a representative sample of households within intervention and comparison areas. In the first stage, a list of all enumeration areas of the study districts was obtained based on the 2007 Ethiopian Housing and Population Census. Two hundred enumeration areas (clusters) were selected with probability proportional to the size of the district. Each cluster served as the primary sampling unit. Within clusters, households were selected by systematic random sampling. The Women's Development Army leaders, HEWs, health posts, health centres with staff and woreda health offices serving the selected clusters were also surveyed.

The sample size was calculated to measure changes with adequate power in a fixed number of percentage points of key indicators between intervention and comparison areas at baseline and endline. Based on the Ethiopian DHS data, a cross-sectional survey of 3000 households in 100 intervention and 100 comparison clusters was expected to find 1747 children under the age of 5 years in each arm.[21] A Tanzanian childhood study found that 50% of under-fives had an illness in the 2 weeks before the survey.[22] The current research assumed more conservatively 30% of children 2–59 months to have had any illness during the 2 weeks before the interview. This sample size of 3000 households per group with 90% completeness and a design effect of 1.3 would give 80% power to detect differences of 10–20 percentage points across a range of child health indicators (5% significance level). The baseline survey found fewer than the expected number of sick children in the 2 weeks before the survey. As a result, the household sample size for the endline survey was doubled. We used the baseline list of households to select 66 households randomly in each enumeration area.

## Measurements

For every selected household, we interviewed the household head, listed residents and collected sociodemographic characteristics. The interviews included caregivers of children aged 2–59 months to assess their knowledge of childhood illnesses and care-seeking for sick under-five children in the 2 weeks before the survey. Furthermore, women of reproductive age (13–49 years) were interviewed to identify births in the 12 months before the survey, with additional questions on care-seeking for illness in the neonatal period. Up to three visits were made to each participant to ensure the study reached its target sample size.

**Table 1** Optimising the Health Extension Programme intervention implemented in 26 districts of Ethiopia, the assumptions and the expected intermediate indicators and outcomes

**Assumptions**

► Health managers and political leaders at all levels will be committed to supporting the intervention.
► Strong coordination and partnership among the stakeholders at all levels.
► Community influencers (religious and traditional leaders) will be change agents in promoting maternal, newborn and child health services.
► Public sector and supply chain partners will ensure drug and service availability.

**Intervention**

| Community engagement | Primary care level capacity building | Ownership and accountability |
|---|---|---|
| ► Health post open house<br>► Group discussions led by Women's Development Army leaders<br>► Engaging schools<br>► Engaging religious and traditional leaders<br>► Engaging agricultural extension workers, to reach male partners<br>► Educational health films<br>► Radio spots<br>► Printed information and education communication materials | **Women's Development Army:**<br>► Level one training<br>► Provision of job aids and tools<br>► Community-based data for decision-making training for Women's Development Army leaders<br>**Health extension workers:**<br>► Gap-filling training and job aids<br>► Supportive supervision<br>► Performance review and clinical mentoring meetings<br>► Community-based data for decision-making training of trainers for health extension workers | ► Advocacy for the integration of community-based newborn care and integrated community case management into planning, budgeting, management and information systems.<br>► District-level annual-based planning.<br>► Management standards for health post opening hours.<br>► Ambulance service for children's referral.<br>► Engage kebele (subdistrict) command post in the efforts.<br>► Community forum.<br>► Establish community feedback mechanisms. |
| **Intermediate indicators** | | |
| Caregivers knowledge, practices and community participation on matters relating to child health | Facility readiness in terms of medicine, equipment and supplies as well as health worker training and supervision necessary to provide child health services | Planning and monitoring of child health services in the district health system (availability of ambulance, community-based newborn care and integrated community case management indicators and standardised operational hours for health posts). |

**Outcomes**

1. Primary outcome
Service utilisation from an appropriate provider for sick children between the ages of 2 and 59 months.
2. Secondary outcomes
Appropriate treatment for diarrhoea and pneumonia.
Care-seeking for infants who were ill in the neonatal period.
Appropriate treatment for possible serious bacterial infection in the neonatal period.

The health facility questionnaires assessed the infrastructure, equipment, supplies and staff available on the day of the survey. Also, data were collected from facility registers on services provided to sick children at health posts and health centres in the 3 months preceding the survey. The health centre staff, HEWs and Women's Development Army leader modules covered their background, knowledge, training in the last 12 months, supervision in the previous 6 months and the services they provided in the last 3 months. Interviewers collected information at the district health office on demography and characteristics that might affect services for under-five children.

The questions and content of each survey module were based on existing large-scale survey tools and the authors' previous evaluation of the iCCM and CBNC programmes.[22][23] All questionnaires were translated into three local languages (Amharic, Oromifa and Tigrigna), pretested and revised. Data collectors and supervisors were trained over 10 days, including field training before the start of data collection. They were not provided information on whether a district was an intervention or comparison area.

Data were collected on personal digital assistants (Companion Touch 8), and tablets (Toshiba and Hewlett Packard) programmed with CSPro 6.3 at baseline and CSPro 7.1 at the endline. Data collectors sent encrypted data from the field to the password-protected server at the Ethiopian Public Health Institute. Data managers conducted quality checks and feedback to field teams. Data were cleaned, checked for errors, including

consistency and completeness. Since OHEP was a community and health system level intervention, a data monitoring committee was not deemed necessary.

## Outcomes

The primary outcome was the proportion of children aged 2–59 months who were reported to have had any illness in the 2 weeks before the survey for whom advice or treatment was sought from an appropriate provider (health post, health centre, hospital and private clinic).

Secondary outcomes included: 1) the proportion of sick children aged 2–59 months who were reported to have received appropriate treatment for diarrhoea (oral rehydration solution (ORS) with or without zinc tablets) and possible pneumonia (antibiotics), 2) the proportion of infants born in the 12 months before the survey who were reported to have had any illness in the first 28 days of life for whom advice or treatment was sought from an appropriate provider and 3) the proportion of infants born in the 12 months before the survey who received adequate treatment for suspected neonatal sepsis (antibiotics). Since malaria was not a common illness in the study areas at the time of the surveys, the assessment of appropriate treatment for this illness was excluded. Additionally, the registers for children aged 0–59 days and 2–59 months were reviewed to assess the median number of children seeking care at health centres and health posts in the 3 months before the surveys.

We also evaluated intermediate indicators that included, at the community level, the proportion of caregivers that knew signs of illness in children and the proportion that cited appropriate action to be taken for a sick child under 5 years of age. We also assessed the proportion of caregivers that reported receiving health messages on common childhood illnesses and those attending community meetings to discuss maternal, newborn and child health issues. At the health system level, we assessed the proportion of HEWs that had received training and the proportion that had attended performance review and clinical mentoring meetings. We also evaluated the proportion of health centres and health posts that had received supervision and the proportion that had the necessary equipment, supplies and drugs for the provision of child health services. District-level ownership and accountability for child health programmes were reflected in the proportion of districts that had iCCM and CBNC scorecards. These cards were programme management tools for setting targets and monitoring performance. Information was also collected on the average number of ambulances available in districts to transport sick under-five children and whether there were standardised hours of operation for health posts.

## Analyses

Descriptive analysis of baseline and endline characteristics in intervention and comparison areas was conducted at the household, caregiver, child, health facility and district levels. Categorical variables were summarised using percentages with 95% CIs. We used means, with SEs, or medians, with IQRs, to summarise continuous variables.

At the district level, the demographic and health system-level characteristics were examined. For households, the characteristics of mothers or caregivers of children aged 2–59 months, and women who had a delivery in the 12 months before the survey were assessed. Distribution of age, religion, education, self-reported distance to the nearest health post and socioeconomic status was analysed in intervention and comparison areas at baseline and endline surveys. Similar assessments were done for the distribution of age and sex among children 2–59 months of age and infants born during the 12 months before the survey. Household socioeconomic status was captured by asset ownership, access to utilities and household characteristics. These were aggregated into a single wealth index score using principal component analysis.[24] The household aggregated scores were grouped into wealth quintiles, where quintile 1 represented the poorest fifth of the households, and quintile 5 represented the least poor fifth. A linear or logistic regression model was fitted, depending on the variable type, to assess if there were any differences between intervention and comparison areas that changed over time. A variable was considered a potential confounder if the differences between intervention and comparison areas showed a statistically significant (p<0.05) change over time.

Caregivers' knowledge, practice and community participation relating to child health were assessed. Furthermore, the health system level factors associated with child health services, including training and supervision of health workers providing under-five services, and the observed availability of infrastructure, equipment, supplies and drugs for the treatment of childhood illnesses at health posts and health centres were assessed.

Using data from facility registers, we also compared the median number of young infants and children 2–59 months of age who received care in the 3 months before the survey in intervention and comparison areas at baseline and endline. We analysed differences between intervention and comparison areas over time using quantile regression analysis.

Difference-in-differences analyses were used to estimate the effect of the OHEP intervention on care-seeking for sick under-five children. Binary outcome indicators were used to capture whether a sick child had sought care or received treatment. The key independent variable for the outcomes of this study was whether the child lived in the OHEP intervention or comparison area. A model was then created that included an interaction term for the timing of the survey (baseline or endline) and the survey area (OHEP intervention or comparison area). This model allowed for the calculation of the odds of care-seeking or treatment for under-five children in intervention areas as compared with comparison areas, accounting for any differences between baseline and endline survey areas, with adjustment for the cluster sampling and identified

confounding factors. The Stata V.13 (StataCorp, College Station, Texas, USA) svy commands were used to adjust for clustering. The assessment used a blinded analysis. The code identifying the intervention and comparison areas were revealed after the analysis and interpretations were completed.

## Patient and public involvement

Patient and/or the public were not involved in the design or conduct, or reporting or dissemination plans of this research.

## RESULTS

The OHEP intervention started in some districts in 2016 and was fully operational from the start of 2017 until October 2018. A process evaluation of the implementation will be reported elsewhere[25] and includes an assessment of implementation fidelity as well as qualitative results on successes and challenges of the implementation. Briefly, a majority of the activities were one-off, assuming that HEWs would catalyse these activities in the communities. The fidelity of the 31 activities varied by district and over time. Many activities were delayed to the last year. In eight of the districts, implementation activities were interrupted 4 months for administrative reasons. Civil unrest in some of the districts had an impact on health services.

Out of the 200 clusters eligible, six intervention clusters were excluded at baseline due to civil unrest and, therefore, not included in the endline survey (figure 2). At endline, a further three intervention clusters and ten comparison clusters were excluded due to civil unrest.

District-level data were not available from five intervention and one comparison districts at baseline, while at endline data were missing from four intervention and two comparison districts (online supplementary table S1). The demographic and health systems characteristics of intervention and comparison districts at baseline and endline surveys were similar. The study districts had an average population size of 130 000 inhabitants, with 23% being women aged 15–49 years, 20% being children below the age of 5 years. The average household size was 4.6 persons. One-third of the districts had a hospital. There were, on average, five health centres per district and five health posts under each health centre. At the time of the baseline survey, there were one to two ambulances available on average in the district for transporting sick children, while this had increased to two to three at the time of the endline survey. There were also some increases in staffing; health officers at the health centres increased from 2 to 2.5, midwives at health centres from 2.5 to >3 and the number of HEWs increased from an average slightly above two per health post to almost 3. These characteristics were similar across intervention and comparison districts. At the time of the baseline survey, CBNC and iCCM indicators were included in the scorecards of 79% of intervention and over 90% of comparison

districts. At the endline, over 90% of all districts had iCCM and CBNC indicators in their scorecards.

The distribution of study participants by age, education and socioeconomic status across intervention and comparison areas showed no evidence of a change over time. There was some evidence that the proportion of Orthodox Christians changed over time, as did the reported distance to the nearest health post. Both were included as confounders in the analysis of the primary outcome. The distribution of sex and age of under-five children remained similar in the study areas over time (table 2). The characteristics of women who had delivered in the 12 months before the survey showed no evidence of change over time. Young infant age distribution changed, which was included as a potential confounder in the analyses (table 3).

There was no association between the intervention and child health messages the caregivers had received or attendance at community health-related meetings (online supplementary table S2). Between baseline and endline, caregivers' unprompted knowledge on sick newborn danger signs, such as limited or no movement and skin pustules, showed some increase in intervention areas as compared with comparison areas over time (online supplementary table S3). At baseline and endline, over four-fifths of the caregivers in intervention as well as comparison areas said they would take their child to the health centre for a range of childhood illnesses, while approximately one-fifth said they would go to a health post (online supplementary table S4).

Two or more HEWs were available in most health posts without association to the intervention (p=0.55 in difference-in-differences analysis) (online supplementary table S5). Most health posts were open 5 days a week without any association with the intervention (p=0.99 in difference-in-differences analysis). The proportion of health posts that posted their operational days and hours decreased in both areas over time.

The proportion of HEWs trained in iCCM and CBNC remained similar at baseline and endline in intervention and comparison areas (table 4). Maternal, newborn and child health orientation for Women's Development Army leaders decreased slightly in intervention areas and had a minor increase in comparison areas (p=0.34 in difference-in-differences analysis).

At baseline, approximately two-thirds of health centre staff in the intervention and comparison areas reported receiving a supervisory visit in the last 3 months; this remained similar at endline (online supplementary table S6). Over half of all the surveyed HEWs at baseline and endline reported receiving a supervisory visit in the last month. Approximately half in both areas and at both time points reported attending a performance review and clinical mentoring meeting in the 6 months before the survey. One in six Women's Development Army leaders reported meeting with HEWs and other leaders in the 3 months prior to the survey in intervention and comparison areas at baseline and remained similar over time.

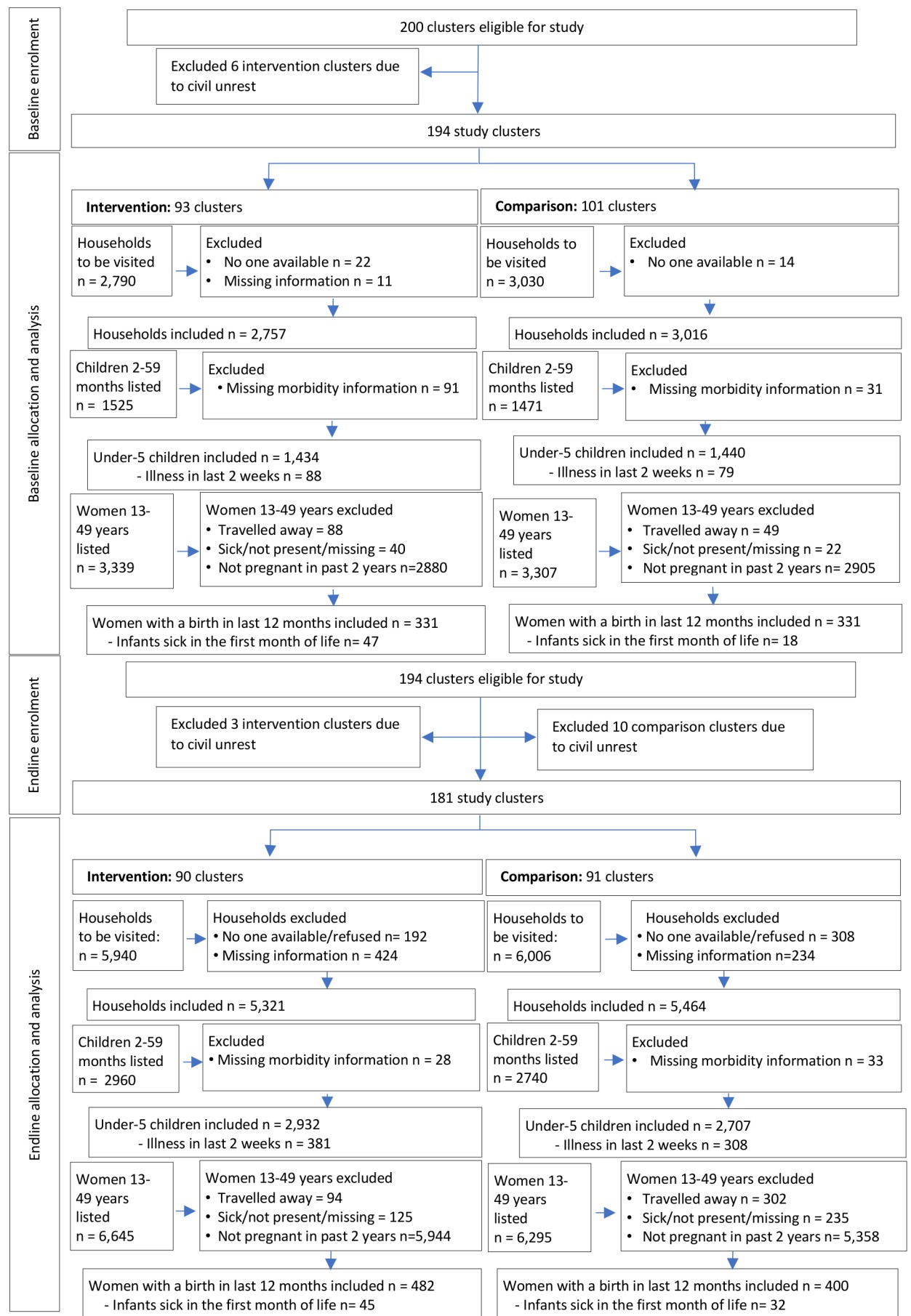

**Figure 2** Study flow diagram of intervention and comparison categorised by baseline and endline data collection period.

**Table 2** Characteristics of mothers or caregivers of children aged 2–59 months and of their children at baseline (December 2016 to February 2017) and endline surveys (December 2018 to February 2019) in intervention and comparison areas

| | Baseline household survey | | | | Endline household survey | | | | |
| | Intervention | | Comparison | | Intervention | | Comparison | | |
| | N | % (95% CI) | N | % (95% CI) | N | % (95% CI) | N | % (95% CI) | P value* |
|---|---|---|---|---|---|---|---|---|---|
| **Caregiver's characteristic** | | | | | | | | | |
| Age (years) | **1259** | | **1273** | | **2454** | | **2275** | | |
| <25 | | 24 (22 to 27) | | 27 (25 to 30) | | 17 (16 to 19) | | 20 (19 to 22) | 0.15† |
| 25 to 29 | | 28 (26 to 31) | | 30 (27 to 33) | | 28 (26 to 29) | | 26 (25 to 28) | |
| 30 to 34 | | 19 (17 to 22) | | 17 (15 to 19) | | 23 (21 to 24) | | 21 (19 to 23) | |
| 35 to 39 | | 16 (14 to 19) | | 16 (14 to 18) | | 19 (17 to 21) | | 18 (16 to 19) | |
| ≥40 | | 12 (10 to 14) | | 10 (8 to 12) | | 13 (12 to 15) | | 14 (13 to 16) | |
| Education | **1259** | | **1273** | | **2454** | | **2275** | | 0.07‡ |
| Schooling | | 36 (34 to 39) | | 45 (43 to 48) | | 37 (35 to 39) | | 45 (42 to 47) | |
| No schooling | | 64 (61 to 66) | | 55 (52 to 57) | | 63 (61 to 65) | | 55 (53 to 58) | |
| Religion | **1259** | | **1273** | | **2454** | | **2275** | | |
| Orthodox Christians | | 54 (51 to 57) | | 38 (36 to 41) | | 61 (59 to 63) | | 47 (45 to 49) | 0.04‡ |
| Others§ | | 46 (43 to 49) | | 62 (59 to 64) | | 38 (37 to 41) | | 53 (51 to 55) | |
| Socioeconomic status | **1259** | | **1273** | | **2454** | | **2275** | | 0.81† |
| Q1 (poorest) | | 19 (17 to 22) | | 23 (21 to 25) | | 18 (17 to 20) | | 25 (23 to 27) | |
| Q2 | | 18 (16 to 21) | | 20 (18 to 22) | | 19 (18 to 20) | | 22 (21 to 25) | |
| Q3 | | 20 (18 to 23) | | 18 (16 to 21) | | 19 (18 to 21) | | 18 (16 to 19) | |
| Q4 | | 22 (20 to 25) | | 21 (18 to 23) | | 20 (19 to 22) | | 17 (16 to 19) | |
| Q5 (least poor) | | 19 (18 to 22) | | 18 (16 to 20) | | 23 (22 to 25) | | 17 (16 to 19) | |
| Distance to the nearest HP | **1104** | | **1126** | | **2030** | | **1878** | | <0.01‡ |
| ≤30 min | | 59 (56 to 62) | | 71 (69 to 74) | | 64 (62 to 66) | | 64 (62 to 67) | |
| >30 min | | 40 (39 to 44) | | 29 (26 to 31) | | 36 (34 to 38) | | 36 (33 to 38) | |
| **Child 2–59 months of age** | | | | | | | | | |
| Sex | **1525** | | **1471** | | **2960** | | **2740** | | |
| Male | | 52 (49 to 54) | | 51 (48 to 54) | | 52 (50 to 53) | | 51 (49 to 52) | 0.77‡ |
| Female | | 48 (46 to 51) | | 49 (46 to 51) | | 48 (47 to 50) | | 49 (48 to 51) | |
| Age (months) | **1488** | | **1453** | | **2960** | | **2740** | | |
| 2 to 11 | | 19 (17 to 21) | | 18 (16 to 20) | | 16 (14 to 17) | | 15 (14 to 16) | 0.43† |
| 12 to 23 | | 19 (17 to 21) | | 20 (18 to 22) | | 18 (17 to 20) | | 16 (15 to 18) | |
| 24 to 35 | | 23 (21 to 25) | | 22 (20 to 24) | | 20 (19 to 21) | | 19 (18 to 20) | |
| 36 to 47 | | 24 (21 to 26) | | 26 (24 to 28) | | 27 (26 to 29) | | 27 (26 to 29) | |
| 48 to 59 | | 16 (14 to 18) | | 14 (13 to 16) | | 19 (17 to 20) | | 22 (21 to 24) | |

*All models were adjusted for clustering.
†P value obtained from linear regression model for the variable to assess whether there was any difference between the groups that changed over time.
‡P value obtained from logistic regression model for the variable to assess whether there was any difference between the groups that changed over time.
§Includes Catholics, Muslims and Protestants.

Almost all health centres and most health posts had patient toilets (table 5). Water availability on the day of the survey decreased over time in intervention area health centres and health posts, and at endline about three-quarters of health centres and half of the health posts in both study areas had water. Electricity on the day of the survey was available in approximately two-thirds of the health centres and one-fifth of the surveyed health posts at baseline and endline.

At baseline and endline, antibiotics needed to treat sick under-five children were available in most health centres (table 5). At health posts, over three-quarters had amoxicillin. However, gentamicin was not available in over half of the intervention and comparison area health posts

**Table 3** Characteristics of mothers and their children born in the 12 months prior to the survey at baseline (December 2016-February 2017) and endline surveys (December 2018-February 2019) in intervention and comparison areas

| | Baseline household survey | | | | Endline household survey | | | | |
| | Intervention | | Comparison | | Intervention | | Comparison | | |
| | N | % (95% CI) | N | % (95% CI) | N | % (95% CI) | N | % (95% CI) | P-value* |
|---|---|---|---|---|---|---|---|---|---|
| **Mother's characteristic** | | | | | | | | | |
| Age (years) | **331** | | **331** | | **482** | | **400** | | |
| <25 | | 34 (29 to 40) | | 34 (29 to 39) | | 25 (21 to 29) | | 29 (25 to 34) | |
| 25 to 29 | | 37 (3 to 43) | | 38 (32 to 43) | | 37 (33 to 42) | | 33 (29 to 38) | 0.73† |
| 30 to 34 | | 16 (13 to 21) | | 15 (12 to 19) | | 22 (19 to 26) | | 22 (18 to 27) | |
| >=35 | | 12 (9 to 16) | | 13 (10 to 17) | | 16 (13 to 20) | | 16 (12 to 20) | |
| Education | **331** | | **331** | | **482** | | **400** | | 0.41‡ |
| Schooling | | 42 (36 to 47) | | 50 (45 to 56) | | 44 (40 to 49) | | 54 (49 to 59) | |
| No schooling | | 58 (53 to 64) | | 50 (44 to 55) | | 56 (51 to 60) | | 46 (41 to 51) | |
| Religion | **331** | | **331** | | **482** | | **400** | | |
| Orthodox Christians | | 47 (41 to 52) | | 32 (27 to 37) | | 57 (52 to 61) | | 41 (36 to 46) | 0.22‡ |
| Others§ | | 53 (48 to 52) | | 68 (63 to 73) | | 43 (39 to 48) | | 59 (54 to 64) | |
| Socio-economic status | **331** | | **331** | | **482** | | **400** | | 0.29† |
| Q1(poorest) | | 21 (17 to 26) | | 21 (17 to 26) | | 21 (17 to 25) | | 29 (25 to 34) | |
| Q2 | | 20 (16 to 25) | | 18 (14 to 23) | | 22 (18 to 26) | | 2 (18 to 26) | |
| Q3 | | 22 (18 to 27) | | 17 (14 to 22) | | 18 (15 to 22) | | 16 (13 to 20) | |
| Q4 | | 17 (13 to 21) | | 21 (16 to 26) | | 19 (15 to 22) | | 19 (11 to 23) | |
| Q5(least poor) | | 20 (16 to 25) | | 22 (18 to 27) | | 21 (17 to 25) | | 14 (13 to 18) | |
| Distance to the nearest HP | **304** | | **303** | | **413** | | **343** | | 0.16‡ |
| <=30 mins | | 60 (54 to 65) | | 69 (64 to 74) | | 61 (56 to 66) | | 66 (61 to 71) | |
| >30 mins | | 40 (35 to 46) | | 31 (26 to 36) | | 39 (34 to 44) | | 34 (29 to 39) | |
| **Child born in the last 12 months** | | | | | | | | | |
| Sex | **324** | | **328** | | **482** | | **399** | | |
| Male | | 49 (43 to 54) | | 52 (46 to 57) | | 55 (50 to 59) | | 52 (47 to 57) | 0.64‡ |
| Female | | 51 (46 to 57) | | 48 (43 to 54) | | 45 (41 to 50) | | 48 (43 to 53) | |
| Age (months) | **331** | | **331** | | **482** | | **400** | | |
| 0 to 2 | | 21 (17 to 26) | | 26 (22 to 31) | | 22 (18 to 25) | | 22 (19 to 27) | 0.02† |
| 3 to 5 | | 24 (20 to 29) | | 32 (27 to 37) | | 25 (21 to 29) | | 24 (20 to 28) | |
| 6 to 8 | | 30 (25 to 35) | | 23 (18 to 28) | | 26 (23 to 30) | | 29 (25 to 33) | |
| 9 to 11 | | 25 (21 to 30) | | 19 (15 to 24) | | 27 (23 to 31) | | 25 (21 to 29) | |

*All models were adjusted for clustering.
†P value obtained from linear regression model for the variable to assess whether there was any difference between the groups that changed over time.
‡P value obtained from logistic regression model for the variable to assess whether there was any difference between the groups that changed over time.
§Includes Catholics, Muslims and Protestants.

at baseline and endline. ORS and zinc were available in almost all health centres at both time points. Zinc was also available in most health posts at both time points. ORS availability increased over time in both intervention and comparison areas. Almost all health centres had job aids and equipment needed for managing sick under-five children. Health posts lacked some equipment in both study areas at baseline and endline, while most had the necessary job aids at both time points.

The median number of sick young infants registered in intervention area health centres in the quarter before the survey at baseline was five and increased to nine at the endline, while the median remained at three for comparison areas health centres (p=0.08) (table 6). Intervention

**Table 4** Child health training for health extension workers and Women's Development Army leaders at baseline (December 2016 to February 2017) and endline surveys (December 2018 to February 2019) in intervention and comparison areas

| | Baseline frontline worker survey | | Endline frontline worker survey | | Difference-in-differences* | |
| | Intervention | Comparison | Intervention | Comparison | | |
| Training | % (95% CI) (n) | % (95% CI) (n) | % (95% CI) (n) | % (95% CI) (n) | % | P value† |
|---|---|---|---|---|---|---|
| iCCM for health extension workers | 83 (74 to 90) (145) | 82 (74 to 88) (131) | 78 (69 to 85) (141) | 80 (70 to 87) (133) | −3 | 0.63 |
| CBNC for health extension workers | 64 (54 to 73) (145) | 66 (56 to 74) (130) | 69 (61 to 76) (141) | 69 (60 to 77) (133) | 2 | 0.89 |
| MNCH‡ orientation for Women's Development Army leaders | 70 (60 to 74) (93) | 59 (48 to 68) (94) | 62 (53 to 70) (169) | 61 (52 to 70) (167) | −10 | 0.34 |

*Difference-in-differences: the difference in the proportion between intervention and comparison areas at endline subtracted from the difference in proportion between intervention and comparison at baseline.
†P value obtained from a logistic regression model for the difference-in-differences analysis.
‡MNCH orientation in the last 12 months.
CBNC, community-based newborn care management training; iCCM, integrated community case management training; MNCH, maternal, newborn and child health.

area health centres at baseline and endline had registered a similar median number of sick children aged 2–59 months (237 vs 232), while comparison area health centres registered fewer (149 vs 128) median number of cases (p=0.58). The median number of sick young infants seen at intervention and comparison area health posts for the quarter before the survey was zero at both baseline and endline. The median number of children aged 2–59 months seen increased from 18 to 22 in intervention areas and decreased from 13 to 10 in comparison areas (p=0.22).

At baseline, 6% of intervention and 5% of comparison area caregivers reported that their children aged 2–59 months had been sick in the 2 weeks before the survey (table 7). At endline, reported illness had increased (13% and 11%). Care-seeking for any illness at baseline in intervention areas was 58% and 49% in comparison areas, while at endline it was 39% in intervention areas and 34% in comparison areas (difference-in-differences −4 percentage points, adjusted OR (AOR)=0.49 (95% CI 0.12 to 1.95)). The use of ORS with zinc for diarrhoea was 32% in the intervention areas and 25% in comparison areas at baseline, while at endline it was to 38% in intervention areas and 30% in comparison areas (difference-in-differences +1, AOR=0.70 (95% CI 0.04 to 13.66)). The use of ORS with or without zinc was higher in intervention than in comparison areas (53% vs 40%) at baseline, while at endline it was similar (50% vs 53%) in both areas (difference-in-differences −16, AOR=0.29 (95% CI 0.02 to 5.33)). Antibiotic treatment for reported signs and symptoms of possible pneumonia in the intervention areas was higher than in comparison areas at baseline (67% vs 56%). At endline, it was 62% in intervention and 69% in comparison areas (difference-in-differences −18, AOR=0.15 (95% CI 0.00 to 16.18)).

Among those that reported illness in the neonatal period, over 70% reported signs and symptoms of possible sepsis (table 8). At baseline care-seeking for any neonatal illness was higher in intervention as compared with comparison areas (74% vs 44%), while at endline it was lower (51% vs 68%) in the intervention than in comparison areas (difference-in-differences −47, AOR=0.04 (95% CI 0.00 to 0.60)). Antibiotic treatment among those with possible sepsis at baseline was 51% in the intervention and 36% in comparison areas and. At endline, it was 57% in the intervention and 68% in comparison areas (difference-in-differences −26, AOR=0.19 (95% CI 0.02 to 2.42)).

## DISCUSSION

This study found that the OHEP intervention neither had any effect on care-seeking for any illness nor on treatment for diarrhoea or possible pneumonia in children 2–59 months of age. Neither did we find an evidence of an effect on care-seeking for neonatal illness nor on the treatment of possible serious bacterial neonatal infection. These findings were based on household surveys and were supported by results from register reviews at health posts and health centres that showed a low level of service utilisation for sick under-five children at baseline and endline surveys. The intervention did not affect caregivers' participation in community engagement activities. No changes were observed in facility preparedness in health centres and health posts that could be related to the intervention. The health system characteristics at the district level showed small changes, which were not associated with the intervention.

The OHEP intervention took place in selected districts of four regions of Ethiopia. Study population characteristics in intervention and comparison districts were broadly similar. A few of the household characteristics differed between intervention and comparison areas but had a marginal influence on the analyses. It is, however, possible that unmeasured confounders might contribute

**Table 5**  Observed availability of infrastructure, equipment, supplies and drugs for treatment of childhood illness at health posts and health centres at baseline (December 2016 to February 2017) and endline surveys (December 2018 to February 2019) in intervention and comparison areas

| | Health centre | | | | | | Health post | | | | | |
|---|---|---|---|---|---|---|---|---|---|---|---|---|
| | Baseline | | Endline | | Difference-in-differences* | P value† | Baseline | | Endline | | Difference-in-differences* | P value† |
| | Intervention n=74 % (95% CI) | Comparison n=81 % (95% CI) | Intervention n=68 % (95% CI) | Comparison n=74 % (95% CI) | % | | Intervention n=84 % (95% CI) | Comparison n=85 % (95% CI) | Intervention n=79 % (95% CI) | Comparison n=86 % (95% CI) | % | |
| **Infrastructure** | | | | | | | | | | | | |
| Patient toilet | 99 (91 to 100) | 96 (89 to 99) | 97 (89 to 99) | 97 (90 to 99) | −3 | 0.51 | 87 (78 to 93) | 80 (70 to 87) | 82 (72 to 89) | 81 (72 to 88) | −6 | 0.54 |
| Water | 91 (82 to 96) | 74 (63 to 82) | 75 (63 to 84) | 69 (57 to 79) | −11 | 0.36 | 69 (58 to 78) | 55 (44 to 66) | 51 (40 to 62) | 52 (42 to 63) | −15 | 0.43 |
| Fridge | 77 (66 to 85) | 78 (67 to 86) | 78 (66 to 86) | 80 (69 to 86) | −1 | 0.838 | 10 (5 to 18) | 9 (10 to 18) | 33 (23 to 44) | 40 (30 to 50) | −8 | 0.027 |
| Steriliser | 70 (57 to 80) | 63 (52 to 73) | 74 (62 to 83) | 68 (59 to 77) | −1 | 0.751 | 10 (5 to 18) | 7 (3 to 15) | 10 (5 to 19) | 10 (5 to 19) | −3 | 0.515 |
| Electricity | 59 (48 to 70) | 62 (51 to 72) | 65 (52 to 75) | 70 (57 to 79) | −2 | 0.588 | 17 (10 to 26) | 19 (12 to 29) | 23 (15 to 34) | 25 (17 to 36) | 0 | 0.392 |
| **Drugs** | | | | | | | | | | | | |
| Amoxicillin‡ | 97 (90 to 99) | 99 (91 to 100) | 97 (89 to 99) | 99 (81 to 100) | 0 | 0.975 | 79 (68 to 86) | 81 (71 to 88) | 73 (62 to 82) | 80 (70 to 87) | −5 | 0.85 |
| Cotrimoxazole | 97 (90 to 99) | 98 (90 to 99) | 94 (85 to 98) | 95 (86 to 98) | 0 | 0.67 | 40 (30 to 51) | 15 (9 to 25) | 8 (3 to 16) | 6 (2 to 13) | −23 | 0.97 |
| Gentamicin§ | 82 (72 to 90) | 81 (71 to 89) | 94 (85 to 98) | 97 (90 to 99) | −4 | 0.07 | 38 (28 to 49) | 41 (31 to 52) | 37 (27 to 48) | 15 (9 to 25) | 25 | <0.01 |
| Ampicillin¶ | 84 (73 to 91) | 90 (81 to 95) | 97 (89 to 99) | 99 (91 to 100) | 4 | 0.31 | | | | | | |
| ORS | 89 (80 to 95) | 80 (70 to 88) | 93 (83 to 97) | 92 (83 to 97) | −8 | 0.16 | 76 (66 to 84) | 58 (47 to 68) | 92 (84 to 97) | 85 (75 to 91) | −11 | 0.22 |
| Zinc | 12 (6 to 22) | 7 (3 to 16) | 16 (9 to 28) | 16 (9 to 27) | −5 | 0.21 | 82 (72 to 88) | 80 (70 to 87) | 84 (73 to 90) | 80 (70 to 87) | 2 | 0.93 |
| Zinc–ORS¶ combined | 74 (63 to 83) | 77 (66 to 85) | 88 (78 to 94) | 68 (56 to 74) | 23 | 0.03 | 10 (5 to 18) | 5 (2 to 12) | 14 (08 to 24) | 9 (4 to 16) | 0 | 0.59 |
| Malaria RDT | 88 (78 to 94) | 77 (66 to 85) | 87 (76 to 93) | 65 (53 to 75) | 11 | 0.23 | 75 (64 to 83) | 48 (37 to 59) | 65 (53 to 74) | 37 (27 to 48) | 1 | 0.56 |
| RUTF | 82 (72 to 90) | 100 | 88 (78 to 94) | 99 (91 to 100) | 7 | | 5 (2 to 12) | 11 (06 to 19) | 47 (36 to 58) | 74 (64 to 83) | −21 | <0.01 |
| **Equipment** | | | | | | | | | | | | |
| Stethoscope | 100 | 100 | 99 (90 to 100) | 100 | −1 | 0.98 | 74 (63 to 82) | 81 (71 to 88) | 72 (61 to 81) | 56 (45 to 66) | 23 | <0.01 |
| MUAC tape | 100 | 99 (91 to 100) | 99 (90 to 100) | 99 (91 to 100) | −1 | | 98 (91 to 99) | 100 | 97 (90 to 99) | 100 | −1 | 0.95 |
| Thermometer | 100 | 98 (90 to 99) | 100 | 99 (91 to 100) | −1 | | 86 (76 to 92) | 87 (78 to 93) | 85 (75 to 91) | 76 (65 to 84) | 10 | 0.11 |
| Infant scale | 96 (88 to 99) | 96 (89 to 99) | 97 (89 to 99) | 99 (91 to 100) | −2 | 0.49 | 80 (70 to 87) | 81 (71 to 88) | 80 (69 to 87) | 77 (66 to 85) | 4 | 0.52 |
| Ambu face mask | 95 (86 to 98) | 94 (86 to 97) | 96 (87 to 99) | 96 (88 to 99) | −1 | 0.69 | 81 (71 to 88) | 71 (60 to 79) | 23 (15 to 34) | 13 (7 to 22) | 0 | |
| Weighing sling | 86 (76 to 93) | 95 (87 to 98) | 93 (83 to 97) | 92 (83 to 97) | 10 | 0.23 | 70 (59 to 79) | 76 (66 to 84) | 75 (64 to 83) | 78 (68 to 86) | 3 | 0.94 |
| Syringes and needle | 100 | 95 (87 to 98) | 100 | 100 | −5 | | 85 (75 to 91) | 81 (71 to 88) | 78 (68 to 86) | 73 (63 to 82) | 1 | 0.55 |
| Functional timer** | | | | | | | 21 (14 to 32) | 22 (15 to 33) | 5 (2 to 13) | 16 (10 to 26) | −10 | 0.38 |
| **Job aids** | | | | | | | | | | | | |
| 0–2 months registration book | 95 (86 to 98) | 93 (84 to 97) | 97 (89 to 99) | 95 (86 to 98) | 0 | 0.98 | 92 (83 to 96) | 88 (79 to 94) | 94 (85 to 97) | 86 (77 to 92) | 4 | 0.53 |
| 2–59 months registration book | 100 | 99 (91 to 100) | 96 (87 to 99) | 100 | −5 | | 95 (88 to 98) | 89 (81 to 94) | 95 (87 to 98) | 94 (87 to 98) | −5 | 0.31 |
| Chart booklet | 96 (88 to 99) | 95 (87 to 98) | 99 (89 to 100) | 95 (86 to 98) | 3 | 0.50 | 87 (78 to 93) | 87 (78 to 93) | 94 (85 to 97) | 90 (81 to 95) | 4 | 0.77 |

Continued

**Table 5** Continued

**Health centre**

| | Baseline Intervention n=74 % (95% CI) | Baseline Comparison n=81 % (95% CI) | Endline Intervention n=68 % (95% CI) | Endline Comparison n=74 % (95% CI) | Difference-in-differences * % | P value† |
|---|---|---|---|---|---|---|
| Supervision checklist†† | 92 (83 to 96) | 93 (84 to 97) | 90 (80 to 95) | 85 (75 to 92) | 6 | 0.28 |
| Family health card‡‡ | | | | | | |

**Health post**

| | Baseline Intervention n=84 % (95% CI) | Baseline Comparison n=85 % (95% CI) | Endline Intervention n=79 % (95% CI) | Endline Comparison n=86 % (95% CI) | Difference-in-differences* % | P value† |
|---|---|---|---|---|---|---|
| Supervision checklist†† | | | | | | |
| Family health card‡‡ | 83 (74 to 90) | 87 (78 to 93) | 81 (71 to 88) | 81 (72 to 88) | 4 | 0.45 |

*Difference-in-differences: the difference in the proportion between intervention and comparison areas at endline subtracted from the difference in proportion between intervention and comparison at baseline.
†P value obtained from a logistic regression model for the difference-in-differences analysis.
‡Amoxicillin 250 mg and 125 mg dispersable tablet and 125 mg/5 mL suspenstion.
§Gentamicin 80 mg/2 mL or 20mg/2 mL for the health centre and 20mg/2mL for health posts.
¶Health posts are not expected to have stocks of ampicillin and was hence not assessed.
**Functional timer not assessed at health centre.
††Supervision checklist was not assessed at health posts as stocks are expected to be kept at health centres.
‡‡Family health card was not assessed at health centres as it is a tool mainly used by health extension workers.
MUAC, middle upper arm circumference; ORS, oral rehydration solution; RDT, rapid diagnostic test; RUTF, ready-to-use therapeutic food.

**Table 6** Median number of newborns (0–2 months) and children (2–59 months) registered in health centres and health posts in the quarter prior to survey at baseline (December 2016 to February 2017) and endline surveys (December 2018 to February 2019) in intervention and comparison areas

**Health centre**

| Age group | Baseline facility survey Intervention n=67 Median (IQR) | Baseline facility survey Comparison n=88 Median (IQR) | Endline facility survey Intervention n=61 Median (IQR) | Endline facility survey Comparison n=81 Median (IQR) | Difference-in-differences* Median | P value |
|---|---|---|---|---|---|---|
| 0 to 59 days | 5 (8) | 3 (6) | 9 (11) | 3 (6) | 4 | 0.08 |
| 2 to 59 months | 237 (309) | 149 (171) | 232 (340) | 128 (119) | 16 | 0.58 |

**Health post**

| Age group | Baseline facility survey Intervention n=74 Median (IQR) | Baseline facility survey Comparison n=95 Median (IQR) | Endline facility survey Intervention n=69 Median (IQR) | Endline facility survey Comparison n=96 Median (IQR) | Difference-in-differences* Median | P value† |
|---|---|---|---|---|---|---|
| 0 to 59 days | 0 (2) | 0 (1) | 0 (3) | 0 (0) | 0 | >0.99 |
| 2 to 59 months | 18 (37) | 13 (25) | 22 (33) | 10 (24) | 7 | 0.22 |

*Difference-in-differences: the difference in the median children seen in the 3 months prior to the survey between intervention and comparison area health facilities at endline subtracted from the difference in the median between intervention and comparison area health facilities at baseline.
†P value obtained from using quantile regression analysis.

**Table 7** Illness, care-seeking and case management in the 2 weeks prior to the survey among children 2–59 months of age reported by caregivers from the household survey at baseline (December 2016 to February 2017) and endline surveys (December 2018 to February 2019) in intervention and comparison areas

| | Baseline household survey | | Endline household survey | | Difference-in-differences* | OR (95% CI) | Adjusted OR (95% CI)† |
|---|---|---|---|---|---|---|---|
| | Intervention % (95% CI) (n) | Comparison % (95% CI) (n) | Intervention % (95% CI) (n) | Comparison % (95% CI) (n) | % | | |
| Illness in the last 2 weeks | 6 (5 to 8) (n=1434) | 5 (4 to 8) (n=1440) | 13 (11 to 15) (n=2932) | 11 (9 to 14) (n=2707) | n/a‡ | n/a‡ | n/a‡ |
| *Among ill* | | | | | | | |
| Care-seeking for any illness§ | 58 (47 to 68) (n=88) | 49 (39 to 60) (n=79) | 39 (32 to 45) (n=381) | 34 (27 to 41) (n=308) | −4 | 0.62 (0.19 to 2.04) | 0.49 (0.12 to 1.95) |
| ORS with zinc for reported diarrhoea | 32 (14 to 56) (n=19) | 25 (10 to 50) (n=20) | 38 (25 to 51) (n=72) | 30 (19 to 42) (n=47) | 1 | 0.74 (0.04 to 12.26) | 0.70 (0.04 to 13.66) |
| ORS with or without zinc for reported diarrhoea | 53 (30 to 74) (n=19) | 40 (18 to 66) (n=20) | 50 (38 to 62) (n=72) | 53 (38 to 68) (n=47) | −16 | 0.32 (0.02 to 4.90) | 0.29 (0.02 to 5.33) |
| Antibiotic for possible pneumonia¶ symptoms | 67 (16 to 95) (n=9) | 56 (22 to 85) (n=9) | 62 (38 to 81) (n=21) | 69 (42 to 87) (n=19) | −18 | 0.29 (0.00 to 26.83) | 0.15 (0.00 to 16.18) |

*Difference-in-differences: the difference in the proportion between intervention and comparison areas at endline subtracted from the difference in proportion between intervention and comparison at baseline.

†Adjusted for religion and distance to nearest health post. Sample size is smaller for adjusted ORs: care-seeking=127 missing data on distance; diarrhoea treatment=23 missing data on distance; antibiotics for possible pneumonia=6 missing data on distance.

‡The difference-in-differences, OR and adjusted OR were not calculated for any illness in the last 2 weeks as this was not an outcome assumed to be influenced by the intervention.

§Care-seeking from hospital, health centre, health post or private clinic.

¶Possible pneumonia cases are those children with reported cough and fast breathing or difficult breathing, but not due to a blocked nose.

**Table 8** Morbidity, care-seeking and management among infants in the neonatal period at baseline (December 2016 to February 2017) and endline surveys (December 2018 to February 2019) in intervention and comparison areas

| | Baseline household survey | | Endline household survey | | Difference-in-differences* | | Adjusted OR (95% CI)† |
|---|---|---|---|---|---|---|---|
| | Intervention % (95% CI) (n) | Comparison % (95% CI) (n) | Intervention % (95% CI) (n) | Comparison % (95% CI) (n) | % | OR (95% CI) | |
| **Among all** | | | | | | | |
| Baby ill in the first month of life | 14 (11 to 19) (n=331) | 5 (3 to 9) (n=331) | 9 (7 to 13) (n=482) | 8 (6 to 11) (n=397) | n/a‡ | n/a‡ | n/a‡ |
| **Among ill** | | | | | | | |
| Prevalence of possible neonatal sepsis§ | 83 (70 to 91) (n=47) | 78 (48 to 93) (n=18) | 73 (59 to 84) (n=45) | 84 (65 to 94) (n=32) | n/a¶ | n/a¶ | n/a¶ |
| Care-seeking for ill baby** | 74 (59 to 86) (n=47) | 44 (23 to 69) (n=18) | 51†† (34 to 68) (n=43) | 68‡‡ (49 to 82) (n=31) | −47 | 0.04 (0.00 to 0.49) | 0.04 (0.00 to 0.60) |
| **Among possible sepsis** | | | | | | | |
| Any antibiotic | 51 (34 to 68) (n=39) | 36 (17 to 60) (n=14) | 57 (38 to 73) (n=30) | 68 (43 to 85) (n=25) | −26 | 0.17 (0.01 to 2.06) | 0.19 (0.02 to 2.42) |

*Difference-in-differences: the difference in the difference in the proportion between intervention and comparison areas at endline subtracted from the difference in proportion between intervention and comparison at baseline.
†Adjusted for age of infant.
‡The difference-in-differences, OR and adjusted OR were not calculated for baby ill in the first month of life as this was not an outcome assumed to be influenced by the intervention.
§Possible neonatal sepsis: fever, unable to suckle/feed difficult/fast breathing, severe chest in-drawing, convulsions or lethargy.
¶The difference-in-difference, OR and adjusted OR were not calculated for prevalence of possible neonatal sepsis as this was not an outcome assumed to be influenced by the intervention.
**Care-seeking outside the home.
††Missing data for two individuals.
‡‡Missing data for one individual.

to the observed results. Given that child morbidity has a seasonal variation, both surveys were done during the same months of the year. At baseline, the overall proportion of reported childhood illness in the last 2 weeks was lower than anticipated, and this was higher at the endline. The difference in the reported proportion of sick children could be due to differences in the interaction and the extent of the probing between data collectors and families when enquiring about childhood illnesses. Field interviewers were, however, blinded to the allocation, that is, whether the district where they collected data was an intervention or comparison area. Any differential reporting of care utilisation between intervention and comparison areas is, therefore, unlikely. Service utilisation records from health posts and health centres supported the household-level findings of no change in service utilisation as a result of the intervention.

The three components of the OHEP intervention were based on an analysis of barriers to child health services utilisation in Ethiopia.[19] As one of the three components, OHEP included several strategies to engage community members, which have been reported to be effective in Ethiopia and elsewhere.[26–28] Under this intervention, OHEP engaged male agricultural extension workers to reach male partners to address the identified barrier of mothers needing permission from family decision-makers to seek care. The health post open house introduced the newborn and child health services available free of charge to the community. This activity addressed the lack of knowledge of the health post services as well as the perceptions of poor-quality care and costs for seeking care provided by the HEWs. Engaging schoolteachers also aimed to increase the awareness of health post services among parents, via their children. Parents' lack of knowledge of danger signs, preference for alternative care and local beliefs that hindered care seeking for ill children were tackled by involving religious and traditional leaders and communicating behaviour change thorough radio spots and dramas, educational films, family health guide (a low literacy pictorial guide to promote maternal, newborn and child health), posters and brochures. This study found that the OHEP intervention neither improved the reach of health messages on treatment for childhood illnesses to caregivers, nor did it influence their community engagement related to maternal, newborn and child health issues. Most caregivers also indicated that they would rather take their sick child to a health centre than to the health post.

The second component of the OHEP intervention aimed to build capacity through training, supportive supervision and performance review and clinical mentoring to address the identified barrier of poor quality of services provided by HEWs. No changes were observed in the iCCM and CBNC training of HEWs that could be related to the intervention. Training alone may not be sufficient to improve healthcare provider's performance.[29] Studies conducted in Ethiopia on supportive supervision and performance review and clinical mentorship meetings

have also shown that these can be effective means of improving services provided by HEWs.[30] The OHEP intervention had no effect on the supervision and the performance review and clinical mentoring meetings provided for HEWs. The quality of iCCM services measured at baseline was low, and it is unlikely to have been improved by the training, supervision and infrequent clinical mentoring supported by OHEP.[18]

It is essential to ensure that the actors who are involved in the provision of childhood services are accountable in the development, financing, implementation, and monitoring of the programmes.[31] The third component of OHEP, district-level ownership and accountability, aimed to address the barrier of health post closure and absence of HEWs by working with the district health offices to standardise and display the health post operational days and hours. Very few health posts displayed this information at the time of the endline survey. Lack of essential drugs and supplies were also identified as barriers and OHEP advocated at the district level to ensure that budget was allocated for their purchase. We found that almost all health centres had ORS, amoxicillin and gentamicin, and nearly all health posts had ORS. Some health posts did not have amoxicillin, and most did not have gentamicin. Compared with another recent Ethiopian study, amoxicillin availability was higher, while the availability of gentamycin was similar.[8] To address the problem of distance, implementers also advocated that district ambulances be used to transport sick children. Although more ambulances were available at the endline, the increase was not linked to the intervention.

In the current study, approximately half of the ill children 2–59 months of age sought care at baseline, and this decreased to one-third at the endline. Other studies have reported similar and even lower levels of iCCM service uptake in Ethiopia.[2 14 32] Our baseline result for care-seeking was higher than expected, while the endline proportion follows the gradually increasing trend seen in the DHS surveys.[2 4 21] Treatment for diarrhoea estimated in this study seemed to follow a plausible upward trend seen across the DHS. The estimated antibiotic treatment for possible pneumonia, which was <10% in 2005 and 2011 DHS, was two-thirds in our study. It should be noted that household surveys may not provide a reliable estimate of treatment for possible pneumonia.[33 34] It is important to consider the context in which OHEP was implemented to understand the outcomes in this evaluation. One plausible explanation for the lack of observed effect may be the relatively short duration of the intervention.[35] In comparison with single or multicomponent intervention, complex social and health systems interventions, which have many sequential activities, should not be expected to result in rapid effects on care utilisation.[36] In some districts, several intervention activities were only fully implemented in the second year, that is, late in the project period. It is also possible that the interaction between different components in a complex intervention within a particular setting might introduce unpredictable effects.[37] Some intervention activities, such as the health

post open house, were also one-off efforts that were unlikely to have a sustained effect on service utilisation. The assumptions under which the OHEP intervention was expected to succeed, for example, community influencers becoming change agents to promote child health services, might not have been met. The lack of drugs, particularly for CBNC services, also indicates that the assumption that the public sector and supply chain partners would ensure drug availability was not fully met. In some districts, implementation was also interrupted for several months due to challenges of reaching administrative agreement between implementing and subcontracting partners. Implementation in other districts was also interrupted due to civil unrest, which made it unsafe for project staff to conduct intervention activities. However, it is important to note that civil unrest was present in a similar number of intervention and comparison clusters. Delays were also caused by difficulties in procuring and supplying behaviour change communication tools. Lastly, the OHEP logic framework developed from the analysis of barriers to child health service utilisation in Ethiopia could have benefited from including a behavioural change theory given that the main outcome, care-seeking, required a change in behaviour.

Future interventions should consider other strategies with evidence of improving child health outcomes. Such strategies might include proactive case detection of ill children by Women's Development Army leaders and HEWs through regular door-to-door home visits.[38] In addition to engaging communities, supporting HEWs through the Women's Development Army leaders can also improve child health outcomes.[39] A linked study has shown that at while two-thirds of Women's Development Army leaders had monthly contact with HEWs, their overall knowledge on maternal, newborn and child health was low.[40] While OHEP aimed to improve the competency of Women's Development Army leaders, this aspect of the intervention was poorly implemented.[25] Improving the Women's Development Army leaders knowledge of danger signs so they can convey health messages regularly to their peers as well as identify and refer ill children to health posts could improve services for under-five children.[41]

## CONCLUSION

This evaluation in four regions of Ethiopia showed that the OHEP intervention did not have an effect on care-seeking for sick under-five children. The lack of effect could be attributed to the relatively short period of OHEP implementation, the nature and unmet assumptions of the intervention and implementation interruption. Future funding schemes need to take into consideration that complex interventions with multiple components, including behaviour change, need a more extended implementation period to measure the effectiveness of the programme. This evaluation is linked to an ongoing process evaluation as well as in-depth substudies that address the Women's Development Army leaders'

promotion of maternal and child health, quality of care provided by HEWs, equity and geographic distribution of service utilisation that can also offer further explanations to the observed lack of effect.[18 40 42 43] Given the overall low care-seeking for childhood illnesses in this study continued efforts are needed to strengthen the primary care services for under-five children.

**Author affiliations**
¹Department of Disease Control, Faculty of Infectious and Tropical Diseases, London School of Hygiene and Tropical Medicine, London, UK
²Health Systems and Reproductive Health Research Directorate, Ethiopian Public Health Institute, Addis Ababa, Ethiopia
³Department of Epidemiology and Biostatistics, Institute of Public Health, College of Medicine and Health Sciences, University of Gondar, Gondar, Ethiopia
⁴Maternal and Child Health Directorate, Ethiopia Ministry of Health, Addis Ababa, Ethiopia
⁵School of Public Health, College of Health Sciences, Mekelle University, Mekelle, Ethiopia
⁶Department of Population and Family Health, Faculty of Public Health, Jimma University, Jimma, Ethiopia
⁷Department of Health Systems and Policy, Institute of Public Health, College of Medicine and Health Sciences, University of Gondar, Gondar, Ethiopia
⁸College of Medicine and Health Sciences, Hawassa University, Hawassa, Ethiopia
⁹Tigray Regional Health Bureau, Mekelle, Ethiopia
¹⁰Department of Health Policy and Management, Jimma University, Jimma, Ethiopia
¹¹Oromia Regional Health Bureau, Addis Ababa, Ethiopia
¹²Southern Nations, Nationalities & Peoples Regional Health Bureau, Hawassa, Ethiopia
¹³Institute of Public Health, College of Medicine and Health Sciences, University of Gondar, Gondar, Ethiopia
¹⁴Amhara Regional Health Bureau, Baher Dar, Ethiopia
¹⁵Department of Human Nutrition, Institute of Public Health, College of Medicine and Health Sciences, University of Gondar, Gondar, Ethiopia

**Acknowledgements** The authors would like to thank Lindsay Mangham-Jefferies for drafting the initial protocol and questionnaires for the surveys. The authors would like to thank the field teams that were involved in the data collection as well as the government official that facilitated the administration of the surveys. The authors would also like to thank the study participants who agreed to generously give their time to participate in the study.

**Contributors** The study was conceived by JS and ETL with inputs from DB, YBO, ADef, AB, AAM, MY, MAW, FW, ADes, FAG, DWD, HB, AT and LP. Contributions to training of data collectors, piloting and supervision during data collection were done by DB, YBO, ADef, LP, ADes, FAG, DWD, HB, ATW, TGetah, TGetac and AH. DW was the data manager. JS, LP, YBO, DB and AD analysed and interpreted the data. DB prepared the first draft of the manuscript with contributions from JS, LP, YBO and AD. All authors read and commented on the manuscript and approved the final version.

**Funding** This project was funded by Bill & Melinda Gates Foundation (OPP1132551).

**Disclaimer** The funder had no role in the study design, collection, management, analysis or interpretation of data.

**Map disclaimer** The depiction of boundaries on this map does not imply the expression of any opinion whatsoever on the part of BMJ (or any member of its group) concerning the legal status of any country, territory, jurisdiction or area or of its authorities. This map is provided without any warranty of any kind, either express or implied.

**Competing interests** None declared.

**Patient consent for publication** Not required.

**Ethics approval** Ethical approvals were obtained from the Ethiopian Public Health Institute (Ethics Ref 613/52) and the London School of Hygiene & Tropical Medicine (Ethics Ref 16117). Information sheets translated into the three local languages were read out to study participants to obtain their written informed consent.

**Provenance and peer review** Not commissioned; externally peer reviewed.

**Data availability statement**  Data are available on reasonable request. Request for data can be made to Della Berhanu (della.berhanu@lshtm.ac.uk). A data sharing committee has been established. All requests will be reviewed by this committee and if granted, data will be shared without any identifiers.

**ORCID iDs**
Della Berhanu http://orcid.org/0000-0002-4984-893X
Atkure Defar http://orcid.org/0000-0001-9435-2135
Dawit Wolde Daka http://orcid.org/0000-0001-5465-6345
Lars Åke Persson http://orcid.org/0000-0003-0710-7954

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
