## [Reviewer comments · BMJ Open]

ARTICLE DETAILS

TITLE (PROVISIONAL)	Does a complex intervention targeting communities, health facilities and district health managers increase the utilisation of community-based child health services? A before and after study in intervention and comparison areas of Ethiopia.
AUTHORS	Berhanu, Della; Okwaraji, Yemisrach; Defar, Atkure; Bekele, Abebe; Lemango, Ephrem Tekle; Medhanyie, Araya; Wordofa, Muluemebet; Yitayal, Mezgebu; W/Gebriel, Fitsum; Desta, Alem; Gebregizabher, Fisseha; Daka, Dawit; Hunduma, Alemayehu; Beyene, Habtamu; Getahun, Tigist; Getachew, Theodros; Woldemariam, Amare; Wolassa, Desta; Persson, Lars; Schellenberg, Joanna

VERSION 1 – REVIEW

REVIEWER	Furaha August Muhimbili University of Health and Allied Sciences
REVIEW RETURNED	21-Jun-2020

GENERAL COMMENTS	This is well written manuscript of an important topic in improving neonatal and child health in developing countries. The complex intervention with multiple components is commendable. Can the authors explain why this design was preferred over cluster randomised trial? This may be an added explanation in the methodology section. Table 7 and 8 It is not clear why the difference in difference is n/a (and there is no legend for this) for example illness in the last two weeks (Table 7) difference in difference appears 1. and illness in the first month of life (Table 8). The authors explained the lack of effect on service utilisation could be the interaction of data collectors and the participants. Could it be also that truly the children did not get sick as the families are following advise such as adhering to vaccinations, good and appropriate infant nutrition? or children are well spaced out due to family planning? Was there any campaign of this nature during the intervention? Did the authors conduct any qualitative study to explain the intervention effects or lack of thereof? This could have added more context into the explanation of the results. Though the authors used a logic framework to design the intervention (which is highly thoughtful and commendable) ,could one of the weakness of the study be that no behavioural change
--

	model such as health belief model or theory of planned behaviour was not included in the design. In my opinion this was important as the outcomes of the study were mainly focused on behaviour change.
--	---

REVIEWER	Henry Perry Johns Hopkins University, United States
REVIEW RETURNED	23-Jun-2020

GENERAL COMMENTS	General comments:  1. This is a well-designed thorough study that addresses an important question. The paper should be published. 2. It is important to publish negative results so that we can try to learn from them. 3. This paper, together with others showing low utilization of HEWs for childhood illness, point to a major defect of the current PHC system in Ethiopia. Continued efforts are needed to solve this problem. One possible approach to try next would be some version of Care Groups (see references below) – on a pilot and experimental basis. 4. The approach used to build local capacity for research in Ethiopia through partnerships with local universities provides added value to the paper. 5. Trying to understand why the complex intervention had no effect is not possible because the paper describing the assessment of the implementation of the project interventions is not available. Qualitative research would also be useful. Hopefully the forthcoming paper on process evaluation will shed light on these issues. 6. A statistician should review the multivariate analyses. Specific comments:  1. It would have been helpful to measure at baseline and endline what the knowledge was of mothers about warning/danger signs for childhood illness for which they should seek care. Efforts to augment this knowledge might have been targeted more intensely in the intervention among the WDA. 2. The paper assumes that all treatments of childhood illness took place at the health post or at the health center. Were HEWs not managing neonatal and child illness when they were out in the community (which is a substantial portion of their time)? 3. The paper states that the intervention ended in October 2018 but the endline survey was conducted from Dec 2018 – Feb 2019. Could this have affected some of the results. 4. The Discussion section is good, but it could be strengthened by:  a. More discussion about why the implementation of the intervention was delayed and interrupted b. What might have been the possible effects of political unrest? Was there more of it in the intervention area? Could political unrest have been a reason that overall care seeking declined at endline? c. What has been learned that can help to improve access to care for sick children in the Ethiopian context? It looks like strengthening the link between the Women’s Development Army and HEWs, giving WDA Volunteers more training on warning signs, and being more proactive in finding children in nearby households who need referral could be promising (or calling the HEW on the phone to come to the house!). d. It would have been helpful to have had some idea of the barriers to child health service utilization that were identified in the
--

	UNICEF paper which is unpublished (ref 19) and which formed the basis for designing the interventions. e. Could cite some of the literature concerning examples in which access to care has improved considerably as a result of frequent home visitation (which was not a part of the intervention here). Several references that might be reviewed for inclusion are:  • Johnson AD, Thiero O, Whidden C, et al. Proactive community case management and child survival in periurban Mali. BMJ Glob Health 2018; 3(2): e000634. • Perry H, Morrow M, Borger S, et al. Care Groups I: An Innovative Community-Based Strategy for Improving Maternal, Neonatal, and Child Health in Resource-Constrained Settings. Global health, science and practice 2015; 3(3): 358-69. • Perry H, Morrow M, Davis T, et al. Care Groups II: A Summary of the Maternal, Neonatal and Child Health Outcomes Achieved in High-mortality, Resource-constrained Settings. Global Health: Science and Practice 2015; 3: 370-81. • Davis TP, Wetzel C, Avilan EH, et al. Reducing child global undernutrition at scale in Sofala Province, Mozambique, using Care Group Volunteers to communicate health messages to mothers. Global Health: Science and Practice 2013; 1(1): 35-51. • Edward A, Ernst P, Taylor C, Becker S, Mazive E, Perry H. Examining the evidence of under-five mortality reduction in a community-based programme in Gaza, Mozambique. Trans R Soc Trop Med Hyg 2007; 101(8): 814-22. • Freeman PA, Schleiff M, Sacks E, Rassekh BM, Gupta S, Perry HB. Comprehensive review of the evidence regarding the effectiveness of community-based primary health care in improving maternal, neonatal and child health: 4. child health findings. Journal of global health 2017; 7(1): 010904. • Lunsford SS, Fatta K, Stover KE, Shrestha R. Supporting close-to-community providers through a community health system approach: case examples from Ethiopia and Tanzania. Human resources for health 2015; 13(1): 12. • Findley SE, Uwemedimo OT, Doctor HV, Green C, Adamu F, Afenyadu GY. Comparison of high- versus low-intensity community health worker intervention to promote newborn and child health in Northern Nigeria. Int J Womens Health 2013; 5: 717-28. f. The published protocol paper (Berhanu 2020) states that the intervention started in 2016 (p. 3, bottom left). This is not what the paper under review states. Why the discrepancy? Minor points  1. P. 5 line 27 – over time instead of overtime 2. At bottom of page 6, the reference cited there “2” does not appear to be the correct one. 3. P. 21 line 27 – any childhood illness rather than any illness. 4. Ref 27 (p. 28) has a misspelling.
--	---

VERSION 1 – AUTHOR RESPONSE

Reviewer: 1

Reviewer Name: Furaha August

Institution and Country: Muhimbili University of Health and Allied Sciences

1. This is well written manuscript of an important topic in improving neonatal and child health in developing countries.

Thank you for indicating the importance of the topic.

2. The complex intervention with multiple components is commendable. Can the authors explain why this design was preferred over cluster randomised trial? This may be an added explanation in the methodology section.

We appreciate this comment. We have clarified in the methods section (page 8, paragraph 1) that the intervention districts were selected by the government of Ethiopia and implementing partners for having both a relatively low utilization of primary child health services and the availability and ability of partners to support implementation. Given this, an evaluation with comparison districts and a before-and-after design was a logical choice.

3. Table 7 and 8

It is not clear why the difference in difference is n/a (and there is no legend for this) for example illness in the last two weeks (Table 7) difference in difference appears 1. and illness in the first month of life (Table 8).

We agree that an explanation was missing. We have now, as a footnote to the tables, indicated that the difference-in-difference, odds ratios and adjusted odds ratios were not calculated for illnesses in first two weeks before the survey for 2-59 month old children, illness in the first month of life or possible neonatal sepsis, as the OHEP intervention would not be expected to influence incidence of illness (page 37 and page 38).

4. The authors explained the lack of effect on service utilisation could be the interaction of data collectors and the participants. Could it be also that truly the children did not get sick as the families are following advise such as adhering to vaccinations, good and appropriate infant nutrition?or children are well spaced out due to family planning? Was there any campaign of this nature during the intervention?

We provide possible explanation for the lack of effect of the intervention on service utilization for sick children on page 24, paragraph 1, which includes too short an implementation period for the intervention, unmet assumptions of the intervention, and implementation interruptions and delays. Our comment about the interaction between data collectors and participants was an explanation, not for the intervention's lack effect on service utilization, but rather for the difference in the proportion of children reported to have been sick at baseline and endline surveys. The higher number of reported illnesses at endline in both comparison and intervention districts could be due to more probing by data collectors, which would have captured more children with relatively minor illnesses as compared to intervention and comparison areas at baseline (page 20, paragraph 2). It is possible that better nutrition and vaccinations at baseline could have resulted in less illnesses in the two weeks prior to the survey, but we are not aware of any change in services that would have led to this. Moreover, as OHEP did not aim to reduce morbidity but rather to increase care seeking for any morbidity, we did not explore these findings further.

5. Did the authors conduct any qualitative study to explain the intervention effects or lack of thereof? This could have added more context into the explanation of the results.

We agree that qualitative studies exploring the intervention effects provide better explanations. A process evaluation paper has just been submitted for peer review (Okwaraji et al.) and includes implementation fidelity as well as qualitative results on successes and challenges of the implementation. We have added this information on page 14, paragraph 3.

6. Though the authors used a logic framework to design the intervention (which is highly thoughtful and commendable), could one of the weaknesses of the study be that no behavioural change model such as health belief model or theory of planned behaviour was not included in the design. In my opinion this was important as the outcomes of the study were mainly focused on behaviour change.

Thank you for this comment. The implementers indeed designed the intervention based on a logic model. We agree that the use of a behavioral change theory could have influenced the composition of the intervention and improved the implementation. We have added this as possible explanation to the lack of effect on page 21, paragraph 1.

Reviewer: 2

Reviewer Name: Henry Perry: Institution and Country

Johns Hopkins University, United States

General comments:

1. This is a well-designed thorough study that addresses an important question. The paper should be published.

Thank you. We appreciate the acknowledgment.

2. It is important to publish negative results so that we can try to learn from them.

We agree that it is important to learn from negative findings.

3. This paper, together with others showing low utilization of HEWs for childhood illness, point to a major defect of the current PHC system in Ethiopia. Continued efforts are needed to solve this problem. One possible approach to try next would be some version of Care Groups (see references below) – on a pilot and experimental basis.

We agree that this is a possible approach. We have indicated this in the discussion on page 25, line 1.

4. The approach used to build local capacity for research in Ethiopia through partnerships with local universities provides added value to the paper.

Thank you.

5. Trying to understand why the complex intervention had no effect is not possible because the paper describing the assessment of the implementation of the project interventions is not available.

Qualitative research would also be useful. Hopefully the forthcoming paper on process evaluation will shed light on these issues.

We agree. We have also added more details regarding the process evaluation on page 14 paragraph 3. The process evaluation report (submitted for peer review) includes a fidelity assessment as well as analysis of qualitative interviews that explore implementation successes and challenges.

6. A statistician should review the multivariate analyses.

Thank you—our analysis was guided and checked by statisticians. We would be happy to receive comments from a statistical reviewer if the editor feels that is appropriate.

Specific comments:

1. It would have been helpful to measure at baseline and endline what the knowledge was of mothers about warning/danger signs for childhood illness for which they should seek care. Efforts to augment this knowledge might have been targeted more intensely in the intervention among the WDA.

We agree that this is important information. In the supplementary Table 3 we show caregivers' unprompted knowledge of newborn danger signs at baseline and endline. Additionally, in supplementary Table 4 we present caregivers' knowledge on actions to be taken when a child under five years is sick (fever, diarrhea, respiratory infection or cough and possible neonatal sepsis) at baseline and endline. The OHEP intervention did include a competency training for WDA leaders to be provided by HEWs using the family health guide (low literacy behavior change communication tool covering danger signs in under five children and actions to be taken for these danger signs) and educational films. However, as will be shown in the process evaluation paper (Okwaraji et al.), the training for HEWs and WDA leaders was poorly implemented, only taking place late in the implementation period for some districts and not taking place at all in other districts.

2. The paper assumes that all treatments of childhood illness took place at the health post or at the health center. Were HEWs not managing neonatal and child illness when they were out in the community (which is a substantial portion of their time)?

We agree that this is important to clarify. We asked if caregivers who reported that their under-five year old child had an illness had sought advice or treatment from any source. For those who said "yes" we asked from where, providing options of: 1) Health post, 2) Health centre, 3) Hospital and 4) Other. For those who said 'Other', we additionally coded those that fell under "appropriate provider" such as private clinic. None of the caregivers had reported that they had been provided care at home by HEWs. A similar procedure was followed for care seeking for newborn illness.

3. The paper states that the intervention ended in October 2018 but the endline survey was conducted from Dec 2018 – Feb 2019. Could this have affected some of the results.

Thank you for the comment. The endline survey was done 2-4 months after the intervention ended, and so a short-term transient effect on care-seeking might have been missed. However, we would expect care-seeking as measured through a household survey to change relatively slowly and to be relatively stable over time. We would therefore argue that more time was needed to see an effect of the intervention on the main outcomes, rather than having missed a transient effect in our survey.

4. The Discussion section is good, but it could be strengthened by:

a. More discussion about why the implementation of the intervention was delayed and interrupted

We agree with this comment. We have added more information on page 24, paragraph 1. We mention that implementation interruptions were due to challenges of reaching administrative agreement between implementing and sub-contracting partners, and civil unrest, which made it unsafe for project staff to conduct intervention activities. Delays were mainly due to difficulties in procuring and supplying behavior change communication tools.

b. What might have been the possible effects of political unrest? Was there more of it in the intervention area? Could political unrest have been a reason that overall care seeking declined at endline?

Thank you for this comment. Although we were not able to monitor civil unrest continuously, the information in the flow diagram (Figure 2) and accompanying text (page 14, paragraph 4) shed some light on this question. At the baseline survey, we excluded 6 intervention clusters due to civil unrest, visiting 93 of the original 99 clusters, while we visited the planned 101 comparison clusters. During

the endline survey, we dropped another 3 intervention clusters and 10 comparison clusters, again due to civil unrest, ending up with 90 intervention and 91 comparison clusters. We have included this information on page 24, paragraph 1. This information suggests that at least at the time of the endline survey, there was similar number of clusters in intervention and comparison areas affected by unrest. The process evaluation paper will also provide further information on this matter.

c. What has been learned that can help to improve access to care for sick children in the Ethiopian context? It looks like strengthening the link between the Women's Development Army and HEWs, giving WDA Volunteers more training on warning signs, and being more proactive in finding children in nearby households who need referral could be promising (or calling the HEW on the phone to come to the house!).

We agree with this comment. We have added more on this matter on page 25, paragraph 2. A linked study, which was recently published, shows that WDA leaders had regular interactions with HEWs (Asheber et al., 2020). However, their knowledge on maternal, newborn, and child health was poor. We have indicated that strategies to improve the WDA leaders' knowledge and their support in identifying ill children for treatment could improve primary child health care services.

d. It would have been helpful to have had some idea of the barriers to child health service utilization that were identified in the UNICEF paper which is unpublished (ref 19) and which formed the basis for designing the interventions.

We agree. In the discussion sections on page 21, paragraph 1, page 21, paragraph 2 and page 22 paragraph 2, which describes the three components of OHEP, we have now added the identified barriers and linked OHEP interventions.

e. Could cite some of the literature concerning examples in which access to care has improved considerably as a result of frequent home visitation (which was not a part of the intervention here). Several references that might be reviewed for inclusion are:

Thank you for this recommendation. We have referred to this on page 25, paragraph 1 and also included some of the references you provided.

- Johnson AD, Thiero O, Whidden C, et al. Proactive community case management and child survival in periurban Mali. *BMJ Glob Health* 2018; 3(2): e000634.
- Perry H, Morrow M, Borger S, et al. Care Groups I: An Innovative Community-Based Strategy for Improving Maternal, Neonatal, and Child Health in Resource-Constrained Settings. *Global health, science and practice* 2015; 3(3): 358-69.
- Perry H, Morrow M, Davis T, et al. Care Groups II: A Summary of the Maternal, Neonatal and Child Health Outcomes Achieved in High-mortality, Resource-constrained Settings. *Global Health: Science and Practice* 2015; 3: 370-81.
- Davis TP, Wetzel C, Avilan EH, et al. Reducing child global undernutrition at scale in Sofala Province, Mozambique, using Care Group Volunteers to communicate health messages to mothers. *Global Health: Science and Practice* 2013; 1(1): 35-51.
- Edward A, Ernst P, Taylor C, Becker S, Mazive E, Perry H. Examining the evidence of under-five mortality reduction in a community-based programme in Gaza, Mozambique. *Trans R Soc Trop Med Hyg* 2007; 101(8): 814-22.
- Freeman PA, Schleiff M, Sacks E, Rassekh BM, Gupta S, Perry HB. Comprehensive review of the evidence regarding the effectiveness of community-based primary health care in improving maternal, neonatal and child health: 4. child health findings. *Journal of global health* 2017; 7(1): 010904.
- Lunsford SS, Fatta K, Stover KE, Shrestha R. Supporting close-to-community providers through a community health system approach: case examples from Ethiopia and Tanzania. *Human resources for health* 2015; 13(1): 12.

• Findley SE, Uwemedimo OT, Doctor HV, Green C, Adamu F, Afenyadu GY. Comparison of high-versus low-intensity community health worker intervention to promote newborn and child health in Northern Nigeria. *Int J Womens Health* 2013; 5: 717-28.

f. The published protocol paper (Berhanu 2020) states that the intervention started in 2016 (p. 3, bottom left). This is not what the paper under review states. Why the discrepancy?

Thank you for indicating this. We have clarified on page 14, paragraph 3, that the intervention started in a few districts in 2016, and was fully operational in all districts at the start of 2017.

Minor points

1. P. 5 line 27 – over time instead of overtime

Thank you. We have changed it so it reads over time.

2. At bottom of page 6, the reference cited there “2” does not appear to be the correct one.

Thank you for this comment. We have corrected that the 2016 Ethiopian Demographic and Health Survey found 46% of children with reported diarrhea in the two weeks prior to the survey had received oral rehydration therapy (page 6, paragraph 4).

3. P. 21 line 27 – any childhood illness rather than any illness.

Thank you. We have made the change on page 20, paragraph 2.

4. Ref 27 (p. 28) has a misspelling.

Thank you. We have fixed it.

Reference

1. Ashebir F, Medhanyie AA, Mulugeta A, Persson LA, Berhanu D. Women’s development group leaders’ promotion of maternal, neonatal and child health care in Ethiopia: a cross-sectional study. *Global Health Action*. 2020;13(1):1748845.

VERSION 2 – REVIEW

REVIEWER	Furaha August Muhimbili University of Health and Allied Sciences
REVIEW RETURNED	29-Jul-2020

GENERAL COMMENTS	The authors have addressed fully my comments.
---

REVIEWER	Henry Perry Johns Hopkins University
REVIEW RETURNED	31-Jul-2020

GENERAL COMMENTS	I have reviewed the authors' responses to the initial set of reviews. I have also reviewed the revised paper. I think the responses are appropriate, and the revised paper should be accepted for publication.
--